# Somatic Narratives about Illness. Biometric Visualization of Diseased and Disabled Bodies in Art and Science Projects

**Ewelina Twardoch-Raś** 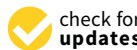

Department of Management and Social Communication, Jagiellonian University, 30-348 Cracow, Poland;
ewelina.twardoch@uj.edu.pl

**Abstract:** This article proposes investigating how the problem of chronic and deadly diseases and bodily injuries is explored in selected contemporary artistic projects based on biometric technologies and medical imaging. All of the projects that will be analysed use specific medical tools and methods (e.g., roentgenography or bio-tracking) to provide detailed, affective images of disability and illness. Nonetheless, these projects were created as pieces of art that combine visual and verbal elements: photographs, collages, and other illustrated stories (e.g., "biometric diaries" or open-source art). On the one hand, they show the "inner" and often invisible face of illness and suffering, but on the other hand they also raise questions related to algorithmic reductionism and politicization of such forms of representation of disease. This article will focus on artistic projects created by Diane Covert, Salvatore Iaconesi and Laurie Frick. It refers to the 'ethos of health' and the conception of ethopolitics (Nicolas Rose) to show the place in contemporary biopolitical society of illness (Thomas Lemke), which can be seen as an exceptional form of the body's condition. Moreover, it considers the problem of the politicization of the biological body and affective experiences (Britta Timm Knudsen and Carsten Stage) and the category of untold histories explored by Joanne Garde-Hansen and Kristyn Gorton.

**Keywords:** illness; health; affect; biopolitics; body; vitality; biometrics

## 1. Biometric Art Practices

The aim of this article is to investigate the representations of bodily vitality, injuries, disabilities and chronic diseases in several artistic projects that combine art—and photography in particular—and science and are based on biometric systems and bioengineering procedures.

Artistic projects based on biological data are difficult to categorize clearly. This difficulty arises from the fact that biometrics is a matter of creative activities implemented in various forms and in different formulas—from designing practices to participatory and immersive environments (Stern 2013). In this regard, art which uses biometric data is no different than most projects in the field of art and science (based on scientific experiments and collaboration between artists and scientists) that have been developed since at least the 1940s and the activity of György Kepes (Wechsler 1978, pp. 7–8). As Stephen Wilson argues, art is now inseparably connected with science, but science also requires increasingly significant social support to develop. Art can help science to be more understandable and thus more socially acceptable (Wilson 2002, pp. 26–30). This is especially important in the case of artistic works focused on illness and disability. In artistic projects in the field of art and science, medical tools and procedures which generally serve diagnostic and therapeutic purposes are often used. Through narrativization and various forms of creative visualisation, especially related to photography, medical

activities are becoming more understandable and engaging for recipients as they often provide a subversive meaning to the procedures and techniques normally used.

Wilson, while describing art projects created using sensors and medical imaging, which he assigns to the general category of information arts, defines them as "technologies that sense internal states of the body" (Wilson 2002, p. 180). I define biometric art projects as those based on procedures for measuring and recording biological data from the surface (e.g., skin) and the inside of the body. Most of the data that forms the basis of the artistic projects that I am interested in are products of various medical imaging methods (X-ray imaging, computed tomography, magnetic resonance imaging, ultrasound, endoscopic examination), functional imaging (functional magnetic resonance imaging), as well as typical biometric techniques and bio-tracking.

Medical imaging techniques are closely related to photography and are considered by many researchers as forms of photographic visualization. First of all, X-ray images for a long time were based on a technique very similar to analog photography (photosensitive plates, photographic film and silver compounds were used), before the arrival of digital procedures (Adrian and Banerjee 2013, p. 74). According to Silvia Casini, medical images are a kind of inside-body photography (Casini 2010). It differs from traditional photography, but it is the only technique that allows to visualize processes occurring in the inside of the body. Lisa Cartwright claims also that "( . . . ) the motion picture, in conjunction with more familiar nineteenth-century medical recording and viewing instruments and techniques, such as the kymograph, the microscope, and the X-ray apparatus, was a crucial instrument in the emergence of a distinctly modernist mode of representation in Western scientific and public culture—a mode geared to the temporal and spatial decomposition and reconfiguration of bodies as dynamic fields of action in need of regulation and control" (Cartwright 1995, p. xi). Medical imaging was thus considered as one of the basic visual techniques of "screening the body", similar to classic photography and motion pictures techniques. What is more, Joanna Zylinska proposes in her book a change in thinking about photography and shows its role as a tool functioning in post-anthropocentric optics. She puts her reflections on visual studies, which she describes as "Nonhuman Photography" and "posthumanist philosophy of photography" (Zylinska 2017, p. 3). Zylinska takes into account CCTV, drones, medical body scans, and satellite images, showing that photography is increasingly separated from human agency and human vision. Medical imaging therefore occupies an important place in contemporary reflection on photography in a philosophical context. Biometrics-based projects that I analyse in the article seem to further develop this reflection.

To be more precise, biometrics and biomedicine means systems and strategies of body measurements and calculations using specific devices (e.g., scanners) to obtain human somatic characteristics, which are often categorized as physiological versus behavioural (Mohamed et al. 2010, pp. 41–56). Behavioural characteristics provide information about, for example, our gait or the rhythm and timbre of our voice. Physiological examples include, but are not limited to, fingerprints, palm veins, face recognition, DNA, palm prints, hand geometry, iris and retina recognition, scent analysis, muscle movement, heart rate, skin conductance response, the functions of particular organs, and changes in tissues and cells (Jain et al. 2011, pp. 3–4). Therefore, on one hand most biometric methods can be used for somatic identification of the individual; however, they can also be used for diagnostic purposes to assess physical condition, health and vitality. In relation to various daily activities, especially those related to the use of mobile devices and social media applications, parameterization methods are also sometimes referred to as "selfie biometrics"—special types of portraits that can be used as an authentication mechanism to better protect our privacy (Rattani et al. 2019, pp. 1–2). Creating "biometric traces" has far-reaching consequences which I analyse more widely in relation to Laurie Frick's artistic projects. Most often, biometric strategies are based on complicated computational procedures and are recorded generally in the form of a graph or diagram (such as data from an electrocardiogram or electroencephalograph). However, in artistic implementations, biometric data are not usually exposed in this form but are automatically transformed using appropriate devices and special algorithms and programs, e.g., into various elements of narratives such as sound effects,

visualizations, and motor activity that stimulates the performer's muscles; they are also combined with other semiotic representations, e.g., language.

The historical roots of this phenomenon can be traced back to Physiognomonics. An ancient Greek treatise on physiognomy was casually attributed to Aristotle as part of the Corpus Aristotelicum. Physiognomy was also developed by other philosophers (Aristotle, Cicero, Seneca the Younger) as a pseudoscientific method that made it possible to designate relations between the shape of the face and the character traits of a man (Popović 2007, pp. 1–7). The idea of Physiognomonics was then developed significantly in the second half of the nineteenth century and at the beginning of the twentieth century as a set of quasi-scientific anatomical identification procedures. Since the nineteenth century, it has been used as an argument in favour of racial divisions and eugenic theories; it has been used to determine personality traits, physiological diseases, and mental pathologies. One of the representatives of physiognomic thought, Timothy Mar, even managed to create a comprehensive typology of various shapes and parameters of the face, such as the structure of the nose, teeth, distribution of wrinkles, etc. (Mar 1974).[1]

Physiognomy is connected with the development of so called "mug shots"—the portrait of a criminal taken after this person was arrested. As explained by Chris Holligan and Dev Maitra, during the late Victorian period, mug shots were the main tool of documenting physiognomy, as well as being very dangerous visual strategies of stigmatisation among the society. At the time, these photographs constituted important archives of images showing people who, due to their physiological features, were predestined to commit crimes. "Criminal anthropology, it is argued, constructed visual sources as tools for reaching certainty, but in this project generated processes of social closure. Morphological deviations from the norm defined the 'criminal body'" (Holligan and Maitra 2018, p. 172). Leesa Rittelmann shows also that photography was one of the main tools used in physiognomics procedures in the Nazi regime. Physiognomy was then strongly related to the way of constructing and evaluating of national identity in Germany (Rittelmann 2010, pp. 137–38). These are only two of many examples of using photography in physiognomics strategies. Photography constituted the documentation of certain anatomical features and it was research material allowing for verification of established quasi-medical concepts. It is not only modern methods of body parameterization that are based on visualization techniques. Photography has been used in them since at least the mid-nineteenth century. For this reason, it is hard to ignore the fact that photography has almost always been one of the basic methods of controlling and disciplining body. It is also important approach to art projects based on biometrics.

According to physiognomics theories human characteristics were reflected in anatomy. Physiognomic methods were used to describe not only abilities but also pathological changes or diseases, especially mental ones. The collected data also helped to highlight racial differences. Notably, physiognomy was an anatomical identification tool that became used in socio-political practices in a similar way as modern complex biometric systems are used for a similar purpose. This resemblance between nineteenth-century physiognomy and modern DNA identification research is indicated by Troy Duster, who analysed the conviction statistics of black people accused of crimes (Duster 2004). Therefore, biological parameterization techniques are inscribed in numerous ideological discourses, such as those concerning racism, ethnicity, gender, various forms of body manipulation for political purposes (eugenics, biopolitics), and other surveillance policies. Modern biometric methods function to a much greater extent within this discourse: on one hand, they create space for the diagnostic

---

1 On the other hand, phrenology, at the turn of the 18th and 19th centuries, made it possible to "determine" the characterological features of a person based on the shape of the skull. According to the findings of Gall and Spurzheim, this was possible by estimating the level of development of the cerebral cortex (Spurzheim 1834). Phrenology influenced the development of neurology and neuropsychology. In turn, pathognomics (the study of facial muscles) is a concept that is still present in psychology, and research on facial muscles is carried out in various fields, including medicine (muscle paralysis, muscle tremors, tics, etc.) (Warren and Philip 2003, p. 74).

practices needed by and useful for patients; on the other hand, they increase the body's involvement in bio-surveillance policies.

Artistic projects based on bio-data incline us to rethink anthropocentric concepts and questions about the agency of non-human actors, namely technological tools and devices for parameterizing body functions. This problem was raised by Rosi Braidotti in her "alliances with technologies" concept (Braidotti 2013), to which I will return to also in another part of this article. Self-tracking tools for instance can be considered as a kind of "supporting technology" that can be used during illness. Artistic projects based on biometric data reflect on the impact of the development of medicine on human and non-human life; this includes not only the problem of the institutionalization of medical practices and the medicalization of life, but also the redefinition of the socio-cultural designation of disease and health. Finally, these artistic works take up socio-political problems that are more and more clearly reflected in the daily lives of individuals. They point to the multifaceted entanglement of biological life in a number of biopolitical mechanisms that are currently mostly economically motivated. Belonging to the domain of engaged art, which simultaneously develops its own strategies and creative tools, they are not servile to certain social, political and autotelic ideas.

This article will look at selected projects created by three artists in the field of biomedical-data art: "Inside Terrorism" by Diane Covert, Salvatore Iaconesi's "La Cura" ("The Cure"), and Laurie Frick's selected works, especially the project titled "Seven stages of ALS". I have selected each of these projects deliberately as each shows different reflections on health and disease and on the role of patients in medicalization processes.

In Covert's project, disability and injuries were caused by a terrorist attack. The body has been inscribed in a specific political discourse, but not to strengthen the ideological tone of the project: the main goal of the work is to reveal the universal, corporeal dimension of the victims of political conflicts. In Iaconesi's artistic experiment, a diseased individual's body is presented from the perspective of resistance against the processes of biopolitics and medicalization, i.e., the institutional regime caused by the development of modern medical practices. Finally, the artistic implementations of Laurie Frick show how modern biomedical and biometric techniques that are used to monitor individuals can be applied to reveal the problems of chronically ill people and to improve their quality of life.

All these projects combine creative visualization techniques (especially photography) and verbal elements. They are implemented through the use of various media, but the relationship between the visual and verbal dimensions is crucial. It is worth adding that this is not usual practice in the field of art and science projects, which generally go beyond verbal forms of representations. All the works discussed here focus primarily on the inside-body dimension of the suffering and discomfort caused by illness and disability. Moreover, in all these projects, biometric techniques subvert their original commercial and institutional function. They convey a kind of critical reflection on the biological parameterization of the body and, at the same time, they provide very specific empathic narratives of injured and diseased individuals.

In the first project, "Inside terrorism", the key is the affective detail which was created to emotionally move the recipient. This understanding of engagement follows the theory of Jill Bennett on experiencing the art in reference to studies of affect (2006). In the second project, "La Cura", a collective forum was created (open-source art) acquires the form of a specific transmedia storytelling which is a recontextualization of Henry Jenkins's category (2006). Laurie Frick's projects explore the possibility of patients gaining self-control over chronic, deadly diseases and of creating an interesting form of solidarity with the patients as agential individuals. This kind of solidarity, according to Btihaj Ajana, is one of the most important phenomena associated with bio-tracking practices (Ajana 2018, pp. 127–28)

By relating the indicated problematics to affective theory and the problem of the mediatization of affective bodily reactions in art, and by considering the problem of the diseased body in relation to reflections related to biopolitics and medicalization processes, I am asking how a patient's status is shaped in relation to biopolitical phenomena. This is a crucial topic in contemporary considerations on the relationship between disease and technology.

## 2. Affective Representations and Biopolitical Strategies

Corporeality in its affective dimension is the basis of all the projects discussed here. These projects are based on biometric data obtained from the body due to the fact that medical imaging studies, biometric procedures, and the sensor-based research used in artistic realizations are all concerned with the biological dimension and reactions of organisms which usually occur without the control of conscious, rational thought. The affective dimension of the body in the selected works constitutes the main material matter of these artistic activities. In this function, affects take the form of a biological body in a material dimension as an element of mediation between tissues, organs, or biochemical processes subjected to research and medical devices. Thus, the levels of affective functioning in projects based on biometric data are complex, heterogeneous and require a multidimensional explication.

In affect theory the most important issues seem to be related to the dichotomous and certainly multi-faceted perception of affect. This dichotomy results from questions such as: to what extent are affects identical to emotions? And: are affects precognitive and pre-individual, and thus do they go beyond the sphere of rational, conscious thought? (Lai et al. 2012; Muncy 1986). Brian Massumi, Nigel Thrift, Teresa Brennan and Patricia Ticineto Clough, have discussed the links between the sphere of emotions and affect. However, they have primarily identified affective reactions of the body with biological bodily (e.g., proprioceptive) experiences. For this reason, an anthropocentric perspective is most often clearly exceeded in their considerations, and corporeality in the affective dimension is here analysed as a sphere of complex biological processes inside the body of human beings (e.g., metabolic processes, cell regeneration), as well as a sphere of connections between human and nonhuman beings (e.g., between people and bacteria). Affects, understood as biological reactions and body states, occur both in human and non-human organisms even if they are not identical (ways of moving of people and plants, hear rates of human beings and animals, etc.). Biological body reactions also enter quite a variety of processes with technologies, such as medical or biometric technologies, therefore they create interconnective links between organic beings, technologies, and sometimes also media as photography (Parikka 2010, pp. 15–17).

This understanding of corporeality, which restores vitality and physiological significance, is also determined by the new relationships that exist between the body and technologies such as various forms of biometric visualization. The development of technology has made it possible to "see" affects and create connections between them and inorganic forms of matter (Clough 2007, p. 2). According to Clough, the affective body is nowadays frequently tied to forms of Deleuzian-understood assemblages between virtual and material elements of reality, and this concept is also conducive to negotiating the boundaries of the organic sphere (Clough 2007, p. 12). By referring to reconfiguration within technology, matter and corporeality, the affective turn also defines the importance of intensifying the processes of the political and social annexation of the body (resulting, among others, from the trends that are creating late capitalism, such as the development of consumer culture), as expressed in various types of biopolitical activities based on control (including self-control), discipline, surveillance, and appropriation of the biological sphere (Clough 2007, pp. 2–3). Although affective experience initially results from the reaction of the organic body, it is also associated with many discursive areas of existence. Therefore, "affect circulation", as Clough calls this tendency, also means the entanglement of biological corporeality as economic value in various production and reproduction systems. In such systems, the body is a significant element of the mechanisms of power in "affective economy" based on "affective labour" (Clough 2007, pp. 25–26).

According to Brennan, the key category that defines the affective turn is "affecting", which is associated with the specificity of psychosomatic experiences. Although it has social and psychological significance, this kind of transmission is founded in the bodily experiences that precede cognitive and discursive mechanisms. Brennan defines this "transmission" as a phenomenon in which emotions and affects—as well as emotional states (such as depression) and the physiological conditions caused by them, such as those associated with illness or disability—can be transmitted between living beings (Brennan 2004, p. 3). This process defines what many theorists call the flow of affect or energy,

but the perspective in which Brennan writes about this phenomenon is derived from her clinical experience: the phenomenon of chemical and energetic "entrainment" that is studied by neurologists and which takes place, for example, due to pheromones or smell in general as well as hormonal interactions that affect the building of emotions and mood at the affective level. Brennan tries to search for scientific explanations for affective phenomena (Brennan 2004, pp. 15–20). Affects have a material and physiological nature and an energy dimension, hence their transmission is possible (Brennan 2004, pp. 24–50). The body is not an entity that is closed to energy or affective exchange with other organisms or the environment, but the lack of clear energy barriers in the body does not entail "levelling" individual affective experiences; on the contrary, affects have both community, relational, and individual forms. Brennan's thesis on energy flow is perfectly confirmed by the projects that I analyse in this article. In particular, the possibility of affective transmission during the perception of art works is discussed in relation to Diane Covert's project.[2]

The body in its affective dimension is very often entangled in various mechanisms of biopolitics, technological regimes and power systems. The biometric methods and data themselves are undoubtedly forms of biological surveillance of the body, as well as biopolitical mechanisms. Medical imaging, biometric methods, and parameterization through sensors are biomedical tools that enable the abstraction and control of biological properties of populations by state authorities and private institutions.

Referring to the findings of Michel Foucault, Thomas Lemke has stated that "the notion of biopolitics refers to the emergence of a specific political knowledge and new disciplines such as statistics, demography, epidemiology, and biology. These disciplines make it possible to analyse processes of life on the level of populations and to "govern" individuals and collectives by practices of correction, exclusion, normalization, discipline, therapeutics, and optimization" (Lemke 2011, p. 5). Management in the dimensions indicated by Lemke is possible, as Foucault emphasized, due to the discovery of the "nature of the population", i.e., statistically recognized biological data which makes a human being belong to the natural world (birth, death, natural or pathological processes of body degeneration, etc.) and which makes it possible—as part of nature—to be subjected to new disciplinary and control procedures (Lemke 2011, p. 6).

Life in this dimension of "new politics" becomes the subject of scientific knowledge, administrative care, and technical optimization. Nowadays, "authoritarian management" no longer functions as a tool for political action but becomes a variety of forms of interventionism, such as stimulating certain actions that optimize, normalize and secure existence. Knowledge systems related to the biological dimension of the population's functioning produce various normative mechanisms that are annexed by the authorities and which grant themselves the right to determine the value or worthlessness of life (in a social and research context), to indicate life-threatening or supporting elements and to define forms of exclusion that are worthy of socio-political attention. Subjectivity perceived in this dimension as a base of biological information becomes the subject of numerous political operations performed in the medical, religious, moral, and scientific fields. The discourse cited by Lemke covers one of the most important issues of biopolitics, which can be generally described as the "ethos of health". As this philosopher claims, "Personal striving for health and wellness is in this way closely allied with political, scientific, medical, and economic interests" (Lemke 2011, p. 102). The phenomenon described by Lemke includes the issues of taking care of health, quality of life, and life expectancy, all of which are possible due to compliance with specific standards and undergoing technical and biochemical procedures (e.g., one-day surgery, aesthetic medicine, vaccination). The productive dimension of life, which results from its optimization and control (in both the governmental and more general public dimension) is, according to this theoretician, one of the most important aspects of contemporary

---

2    To develop this issue further, in the next part of the article I also apply Jill Bennett's theory of affective perception of art works (including the concept of "somatic detail").

biopolitical activities ([Lemke 2011](), p. 36). The difference between past and current practices in this area is expressed in what Lemke points out: in decentralizing power and diverting interest in health into the fields of economy and morality. In the area of economy, we can also add fashion, lifestyle, and fitness. It is quite obvious: someone who is in good shape is healthier and therefore more efficient at work, more disciplined, organized, and productive.

The body's affective sphere is increasingly dependent on complex multimodal technological systems: corporality combined with technological solutions that create hybrid spaces of interconnections. Biometric art projects problematize this issue, first of all by referring to the changing discourse on the problems of health and disease. For this reason, in my analysis of the selected projects I refer to both affective theory and biopolitics theory. The "vulnerable body" concept (the body that is a tool of biopolitical practices) introduced by Britta Timm Knudsen and Carsten Stage is an extension of both these perspectives. I use this category of the "vulnerable body" to analyse Diane Covert's "Inside terrorism" photographic project. I also introduce the category of ethopolitics developed by Nikolas Rose to describe biopolitical phenomena focused on individual management in relation to the project "La Cura" by Salvatore Iaconesi in cooperation with Oriana Persico.

Another variant of shaping somatic and biomedical identities, in the context of health and disease, are artistic projects based on self-tracking methods. These implementations have only been created in recent years and explore the special case of parametrization activities related to the idea of activism for physiological self-improvement and self-control. Self-parameterization, which is the essence of the currently developing ethos of health, is in this variant the transfer of methods of tracking biological life in the sphere of daily practices and activities, and it is also related to chronic illness. Together with the method of external identification and verification of identity, it becomes a technique of gaining self-knowledge and self-optimization in reference to the intimate dimension of the body. Thus, it functions as a more precise and more sophisticated method of controlling biologically understood corporeality, which in a technologically mediated form is an unprecedented phenomenon.

Methods of tracking vital functions and thus parameterizing them with the use of easily available (primarily mobile) tools introduce new perspectives into the area of biopolitical reflection. First of all, they are associated with universal computation: the intense and automated participation of non-human actors. Btihaj Ajana sees self-tracking as the culmination of the development of data-sharing culture ([Ajana 2018](), pp. 8–9). However, this phenomenon implies additional instability and ambiguity in the area of activities related to biopolitics because, at least by definition, it begins to affect the individual space and permeate into the processes of taking conscious control of the affective sphere of corporality. Self-tracking, a variant of tracking procedures that are centred around self-parameterization for the purposes of self-improvement and self-optimization ([Swan 2013](), pp. 85–86), can be read multidimensionally: as an attempt to regain control over the body beyond medical institutions, or as relatively new forms of late-capitalist demands for affective productivity. Repressive institutionalization is still important as a reference point in the categorization of less explicit biopolitical phenomena such as the micropolitics of pharmaceutical and cosmetic concerns. According to Gina Neff and Dawn Nafus, self-tracking seems to ideally implement the scenario of the development of modern biopolitical mechanisms that focus on population management, but starting with the parametrization of individuals' bodies. The main concern also relates to the dangers of marginalization, exclusion, and discrimination that may arise as a result of the obtained data ([Neff and Nafus 2016](), p. 2). As I have pointed out, the discourse regarding various forms of disciplining the body and the mechanisms of social exclusion policy has accompanied bio-parameterization procedures since practices for measuring biological body parameters first became available. This is also part of all modern medicalization processes.

Self-trackers are wearable technology or technologies that you can "wear", which is why they basically fall into the field of mobile media. Self-tracking methods rely on collecting data from living

organisms through various sensors and accelerators built into these devices.[3] These devices and the practices that are created around them constitute a real extension of the medical sphere into everyday private activities (Nafus 2016, pp. 18–19). What is more, many people use self-trackers due to inaccessible healthcare and medical administration systems (Swan 2013, p. 92). As emphasized by Nafus, computational methods based on digital processes have been used in shaping medical knowledge since at least the 1960s (initially, they were genetic engineering experiments), although never before have technologies based on sensors played such a significant role in spreading knowledge about health and illness (Nafus 2016, pp. 15–16).

In her concept of a transversal alliance with technologies, Rosi Braidotti deals with these kinds of technical-organic connections, seeing them as one of the challenges posed in the area of posthumanist ethics. Braidotti derives her concept from reflections on the intimate relationship between human beings and "technological" agents which is realized in the processes of "becoming-machine", which is one of the dimensions of post-human existence. According to Braidotti, this extraordinary connection between human beings and technical entities results from the gradual abolition of the boundaries between what is organic and inorganic, and in the case of electronic devices (such as self-trackers) between electrical circuits and organic nervous systems (Braidotti 2013, p. 89). Braidotti also notices the biopolitical implications of relations with technology and defines these implications as a "cruel" political economy because nowadays every technical mediation of the body carries a number of biopolitical implications. I will develop this issue later.

It is hard not to notice that the role of self-tracking in health care systems is a partially utopian vision. Self-tracking applications help chronically ill people maintain healthy habits and monitor their vital signs. However, they are not a substitute for thorough medical diagnostics, and the fact that bio-tracking is considered as an alternative shows the desperation of patients and the weakness of medical care on a global scale. What is more, as Btihaj Ajana has observed, "The speculative benefits of self-tracking technologies and practices have also been coupled with growing concerns, ranging from privacy, surveillance and data ownership issues to concerns about excessive self-involvement and the pressure of self-improvement that often underscores the practices and ethos of self-tracking culture" (Ajana 2018, p. 4). The other concerns relate to the pressure to increasing the vitality of the body, which should always be attractive and efficient. Based on this principle, the stories of chronic, deadly diseases and disabilities that are told in artistic projects through the use of self-tracking techniques are to some extent a negation of the original idea of contemporary auto-monitoring. After all, these are stories about imperfect, inefficient bodies as opposed to the idea of the "ethos of health". Works of art based on bio-tracking often use applied technologies in a subversive manner by showing the experiences of excluded and marginalized individuals as agential beings. In this way they consider biological parameterization strategies in a more nuanced way, exposing the creative processing and recontextualization of the body's control and management methods, as I will discuss in relation to the works of Laurie Frick.

## 3. Affective Investigation of A Vulnerable Body: Diane Covert's "Inside Terrorism" Project

Referring to the presented conceptualisation of the category of affect, we can consider "the affective" from two perspectives: firstly, as a matter of projects based on biometric data in the biological, intra-body dimension that relate to affective reactions and processes (work of tissues, organs); secondly, as an affective experience induced in the recipient, here especially in relation to photography. Although

---

[3]    In the form of bracelets, smartwatches or, to a limited extent, smartphones, self-trackers measure the number of calories consumed, steps taken, heart rate, pressure, quality and length of sleep, blood sugar, hormone levels, and stress; they manage pregnancy and help observe chronic and incurable diseases. These are only some of the activities that bio-tracking devices can monitor (Swan 2013, p. 87). However, it should be remembered that these devices have numerous limitations: many of the metabolic processes that occur in the body cannot be measured by the sensors installed in ordinary mobile phones. Based on sensors, bio-trackers offer relatively easy and inexpensive access to many biomedical techniques which until recently were only available as a part of the medical care offered by clinical institutions, clinics or private doctors' offices.

it seems to be crucial for bioartistic projects, this basic distinction results from consideration on the role of affect in experiencing the art as introduced by Jill Bennett.[4]

Bennett positions her reflections on trauma, understood as an experience that appears not only as the somatic reactions of a potential recipient of art who physically recreates this trauma to some extent in their own body, but which can also be felt as authentic when it is mediated, for example through a language-based or visual medium (Bennett 2005, pp. 34–37). According to Bennett, trauma, including trauma related to bodily mutilation or disease, has an affective value. Through transmission, affect exists in the relationship—in the flow—between the "content" of the work in which the affective component already exists and the affective way in which the recipient experiences it. Reactions to affective stimuli are experienced somatically; they are caused by physiological processes, and this validates the authenticity of the experience. So, it is not just a feeling of distanced empathy towards someone, it is also commitment to someone's experience that reaches the viewer through their sensual sphere and proprioceptive impressions. In this regard, Bennett refers to Deleuze's understanding of affect as an experience that escapes the will of the viewer and that is not only a conceptual reflection on bodily sensations (Bennett 2005, pp. 7–8). Affect emerges here in two ways: by being affected by someone's affective experiences, which later on are experienced by us, and through our own sensations that have been evoked.

The author refers primarily to visual representations of intense affective experiences such as trauma or violence. The basic issue that structures her considerations is the establishment of a specific relationship that connects the body and mind with visual signs and cannot be reduced to a simple form of representation. Images like this not only evoke affective experiences with the help of references, but also activate and perform them. Bennett states that:

> "The conveyance of suffering through imagery in this context is possible only insofar as images have the capacity to address the spectator's own bodily memory; to touch the viewer who feels rather than simply sees the event, drawn into the image through the process of affective contagion ( . . . ) Bodily response thus precedes the inscription of narrative, of moral emotion or empathy". (Bennett 2005, p. 36)

The category of "affective contagion"—that Bennett borrowed from Silvan Tomkins (2008)—implies a kind of visual affective message that activates somatic reactions in the recipient. This is possible, among others, due to bodily memory, which enables identification on the physiological level with the bodily dimension of the experiences of other beings. This concept also relates to the theory of affect transmission (as already mentioned) investigated by Brennan. However, in contrast to Brennan, Bennett also applies it to the sphere of mediated feelings and reactions. Bennett's concept, which assumes the circulation of affects between an artistic project and the receiving experience, seems to capture the dimensions of affinity that exist in projects based on body-derived data which convey affective experiences through media techniques (semiotic representations) in mediation with non-human factors (Bennett 2005, p. 75).

An interesting example of a project which includes a special kind of affective body dimension marked by mutilation, physical dysfunction, and disease is "Inside Terrorism", which was implemented in 2006–2012 by Diane Covert in the form of verbal and visual collages created on the basis of a series of medical images (http://www.x-rayproject.org). Pictures from X-ray and tomographic examinations were combined with short texts such as symbolic descriptions of bodily injuries, as well as situations where the victims were injured. The somatic photographic collages that Covert creates are narratives about the suffering, sick body. As Michael Heitkemper-Yates claims, "collage reading not only entails a decoding of the visual/verbal/gestural languages of the text, it also necessitates active engagement with

---

[4]   There is no room here to discuss these theories in detail. My aim is to indicate the role Bennett attributes to affect in reference to her artistic practices and perception of art.

the visual/verbal/gestural patterns that constitute the text's grammar and unite to form (or suggest) the syntax of the message (s) communicated by the collage" (Heitkemper-Yates 2015, p. 313). In the case of the Covert project, we are dealing with a special combination of language and visual dimensions obtained by registering data downloaded from the source medium, which is the body.

As they are medical documentation of the injuries sustained by victims of terrorist bomb attacks, all of the collages in Covert's project illustrate specific situations involving the physical dimension of human existence. According to this artist, these pictures constitute actual medical documentation that was collected from the two largest hospitals in Jerusalem and they concern "accidental" victims—civilians who were at the scene of explosions. The photos and scans are from people of different racial, ethnic and religious backgrounds; this is important because—as the artist claims—in the bodily dimension the social differences between victims cease to matter as all their injuries are equally severe and tangible. It is worth mentioning here that this is an interpretation that opposes thinking about biomedical technologies in biopolitical categories, which, as I have already pointed out, leads to social exclusion by introducing the normalization regime. The affective sphere of corporality is presented here as a space of equality and solidarity in suffering, thus in quite a subversive way. What is more, in many of these collages the explosions themselves were symbolically described from the perspectives of the victims; they have inscriptions such as "I was eating pizza" and "I was riding the bus", describing in a concise way everyday situations that turned into a tragedy due to unexpected attacks. Although they remain anonymous in the pictures, with the help of captions the victims are also given a voice and are presented in the project as individuals. In other works, the artist used partially factual descriptions of the injuries that generally appear in these medical images that additionally emphasize their somatic character, such as "Nail in the nape", "Screw in the bone", "Crushed hand", "Head injury", "Injuries of children", and "Broken hand" (http://www.x-rayproject.org). It is worth emphasizing, however, that none of these descriptions are formal, hermetic, or medical: no descriptions are used in medical imaging procedures. Instead, they function as contextual descriptions of the situation and signatures of somatic states of the body that a viewer without medical education would most likely be able to determine himself (serious broken bones can be easily seen in medical photos). The whole project consists of dozens of medical images, each of which documents the physiological effects of attacks.

It seems that all these photos and collages can be seen as "untold histories", in accordance with the approach proposed by Garde-Hansen and Kristyn (2013, p. 12). These stories, as a part of the affective economy, relate here to hidden dimensions of life and to oppression; they reveal the biological processes of injured bodies that for some reason were not publicly disclosed, but due to technology these stories have a chance to be told in a different way. The medical pictures from the Covert project relate to alternatively told stories related to terrorism that in this case were exposed and revealed through suffering and dead bodies and the use of verbal–visual (photographic) collages. On the one hand, referring to these universal representations of corporeality in the form of reliable medical documentation makes this story seem anonymous and undifferentiated: it presents terrorism as a phenomenon strongly associated with socio-political regulations in the biological dimension of all living beings. On the other hand, however, showing these bombings from an affective perspective makes those images extremely intimate and individual because the history of victims is rarely transmitted in such a direct and detailed way. The multimodal narrative structure of the collage allows viewers to be engaged in a somatic spectacle by a multi-faceted activation of their perceptual abilities and by the process of deeper identification with the body representation. These narrative structures are the trace of individual stories that are hidden behind every biomedical and biometric procedure. Contrary to beliefs about the non-narrative status of affects, the language dimension of the Covert project strengthens its affective message and recontextualizes the "somatic detail" of these medical images. Linking verbal and non-verbal narratives is therefore crucial to the idea of the project.

The project's affectivity is achieved in the dimensions Bennett indicates in his concept of "affecting contagion" through visual art projects. So, first of all, the collages in Covert's project can be classified

as affective because the very nature of medical photos, which always show bodily experiences, in this project is intensified by capturing the body in situations of particularly intense pain that is reinforced by photo inscriptions. These are the affective experiences of a truly suffering body that gained its visual–linguistic representations also through the use of medical devices. Affectivity in the receiving dimension, however, also results from somatic reactions to these affective performances, which—following Bennett's line of reflection—manifest themselves not through emphatic observation but through bodily compassion, namely through experiencing "forwarded" pain and suffering. "Affection contagion" here involves a special circulation of affective experiences that allow (of course, not fully due to the unimaginable or "imperceptibility" of this kind of pain) viewers to be affected bodily by photographic representation and, in this case, in the form of medical images.

Bennett's findings also confirm Lisa Blackman's reflection on the mediation that occurs between the body (suffering, dying), even if it is represented in the media rather than directly experienced, and the viewer's ability to produce a somatic response to this kind of bodily experience (Blackman 2012, pp. 76–79). Affects are not simulated here but staged through the use of certain visual and linguistic strategies that make the affects of others accessible to us, despite the complex formula in which they are included by the artists. In this regard, above all it seems important to expose the affective detail/wound which, according to Bennett, causes our extremely intense somatic reactions (Bennett 2001, pp. 9–10). Although the photographs present only the body in its affective, biological dimension, the detail in the project seems to be primarily those fragments of pictures that illustrate the contact of organic tissue with mechanical objects, as is emphasized by their descriptions: a nail punched into cervical vertebrae, shrapnel stuck in bones and muscles, metal fragments cutting tendons. The suggestibility of these physiological processes is probably even greater because—as mentioned by Bennett—it activates the bodily memory: mechanical mutilation of the body is usually such an intense experience that the accompanying pain is re-activated to some extent when it is evoked by images of similar sensations. The role of affective detail is also played by representations of injuries which strengthen the message of the project through the use of tautology.

As I mentioned before, in reference to the affective turn that relates to biopolitical optics, an interesting theoretical proposition which I would also like to use to analyse the Covert project is the "vulnerable body" concept proposed by Britta Timm Knudsen and Carsten Stage. The initial assumption of these researchers is that a sensitive, fragile body always functions as an affective sphere entangled in political mechanisms. This is why Knudsen and Stage emphasize that the affective dimension of corporeality, the sensitive body, is becoming at least an ambiguous "weapon" in the area of political activities that are mediated by global media. On the one hand, due to the phenomenon of "affecting contagion" or "affecting transmission"—not only but also through mediated forms, as photography—the body gives rise to affective reactions from recipients (compassion, indignation) that often become a real spiritus movens of socio-political involvement and change. Affect in this perspective is often presented as the most democratic tool available to ordinary citizens in the area of socio-economic micro-policies (Knudsen and Stage 2015, pp. 2–3). In fact, it is hard to disagree that it is the vulnerable body that is sometimes the bargaining chip in protests against warfare; it influences public awareness of socio-political processes that violate the rights of individuals, especially while dealing with the problem of the victims of political actions—as in Covert's project. In terms of emotional and affective imagination, the body influences the formation of beliefs and shapes the message related to particular political actions. Therefore, the verbal-visual iconography of "vulnerable bodies" in public space is a significant example of the public–private boundary being crossed in discussions about illness and corporal injuries.

On the other hand, in areas where it functions as an important political force, a "vulnerable body" can also become a tool for political and media manipulation. Often, this even happens from the bottom-up perspective; therefore, due to individual activities, biopolitical processes that occur at the state level are legitimized. To justify this thesis, researchers refer to photos posted on social media which show the torture performed by Western soldiers on Iraqi prisoners. These photos are particularly

problematic for the authors as they express support for the uncompromising policy of Western states that use representations of sensitive biological bodies to achieve their political purposes. Therefore, such affective representations can initiate social opposition to given policies and real transformations among the population, but in parallel these activities can be also economically motivated by various political mechanisms.

Covert's project explores this dual dimension of political involvement in reference to one of the most important and most controversial political problems related to globalization and migration policies. All the X-rays scans used are from the two largest hospitals in Jerusalem, therefore they may be considered as a sort of statement on the political situation of Israel, namely the Jewish–Arab conflict. Covert, who is of Jewish origin, suggests that she explores anti-Semitic mechanisms in her project (https://www.x-rayproject.org/), which implies that the pictures can be interpreted as supporting Israel in this conflict by pointing to the harm suffered by Jewish inhabitants as a result of the actions of their antagonists. In this context, a medical picture can function as an individual, intimate picture of a victim's injuries, but it can also be used as an ideological message that justifies and strengthens specific theses and beliefs about a political conflict. This fact confirms Knudsen's and Carsten's considerations that images of a "vulnerable body", especially when created with biomedical technologies, always raise doubts and can be a tool for ideological manipulation. However, this is exactly the result that Covert wanted to avoid. Using X-ray scans, "internal photos", instead of traditional photography, she tried to give her work a universal character, showing the tragedy of the conflict in Israel regardless of its political and cultural context as the victims span a broad range of ethnicities and religions (https://www.x-rayproject.org/). Using medical technology, she tries to give the project a subversive meaning as a tool to explore and reveal the real and universal human suffering that also occurs in the area of the political management of the health and life of citizens. Regardless of the different possible interpretations of the project, Covert's work is an unprecedented artistic voice regarding the universal inside-body aspect of the role of victims in (bio)political conflicts.

This particularly important topic of popularizing knowledge about illness and suffering through art combined with science and technology returns in Salvatore Iaconesi's and Laurie Fricks's projects.

## 4. Salvatore Iaconesi's "La Cura" Project and Ethopolitics

Ethopolitics is a variant of the biopolitical mechanisms. Nikolas Rose derives the concept of ethopolitics from Foucault's assumptions on modern political systems' exploitation of the vitality of life. As Rose emphasizes, individuals have nowadays become political entities whose lives should be managed, optimized, controlled, and estimated at the level of population. According to Rose, biopolitics is currently developing in three directions: risk policy, molecular policy, and the aforementioned ethopolitics (Rose 2007, pp. 41–48). This means the development of new ways of assessing life, determining usefulness and productivity, and creating further forms of biological control of societies (Rose 2007, pp. 161–63). This system is developed in various biopolitical strategies: genetic determinism, the processes of instilling healthy and moral living habits, reproduction control (forced sterilization on medical, social and ideological grounds also occurred in democratic Western countries long after the war), as well as some factors associated with preventive medicine and health education. As Rose states:

> "Today, however, the rationale for political interest in the health of the population is no longer framed in terms of the consequences of population unfitness as an organic whole for the struggle between nations. Instead it is posed in economic terms, the costs of ill health in terms of days lost from work or rising insurance contributions, or moral terms, the imperative to reduce inequalities in health". (Rose 2007, p. 63)

In accordance with the assumptions of ethopolitics, the new subjects of political activities (which in the context of Rose's considerations are primarily individuals) are forced to take responsibility for their own health because they must meet social norms. They become actors and consumers who through various methods (e.g., alternative medicine or medical activities such as various forms of

self-monitoring) increase the vitality and capabilities of their bodies. Thus, they function as individual links in ethical and political regimes (Rose 2007, p. 23).

Rose also classifies new identities that emerge at the individual level as "somatic", hence identities centred around the idea of the vitality of the body are most often socially programmed through various counsellors and institutions in order to develop and improve corporality (Rose 2007, pp. 25–26). As Rose claims:

> "If 'discipline' individualizes and normalizes, and 'biopolitics' collectivizes and socializes, 'ethopolitics' concerns itself with the self-techniques by which human beings should judge and act upon themselves to make themselves better than they are ( … ). This biological ethopolitics—the politics of how we should conduct ourselves appropriately in relation to ourselves, and in our responsibilities for the future—forms the milieu within which novel forms of authority are taking shape". (Rose 2007, p. 27)

In this respect, private life functions as a value related to normalized biological social standards. Biomedicine began to perform the function of not only preventing death but also increasing the efficiency of life. It should strive to optimize corporeality and cover general <<welfare>>: beauty, success, happiness, sexuality, and many others. Somatic individuals understood in this way (or in this "somatic agency" context, using the terms introduced in post-anthropocentric approaches) are the main actors of ethopolitics (Rose and Abi-Rach 2013, pp. 21–23).

According to Rose, both the new types of individual responsibilities as well as individual alienation, exclusion, and social asymmetries are formed on the basis of biological efficiency and productivity. In addition, through this type of conditioning a new type of "vitality normativity" is being developed which determines the possible directions of functioning of human beings independently of the privileges of sovereign power (Rose 2007, p. 26). In relation to this type of normalization, the vitality of individuals is assessed: ability to work, reproductive capacity, propensity for illness, and degree of physical and mental disability. This creates a huge area of vital exploitation that is closely related to the development of medicalization processes, especially on the level of the micropolitics of medical and pharmaceutical concerns. Individuals have become political entities whose lives should be managed, optimized, controlled in terms of the population, and these processes particularly affect disabled people or individuals who, as a result of illness, are temporarily or permanently excluded from work and require constant health care. Their intimate life then becomes the object of administrative management and medical control, under which they lose their agency.

Artistic reflection on the biomedical dimensions of the functioning of biopolitics that corresponds with Rose's considerations is developed in the project "La Cura" by the Italian artist Salvatore Iaconesi with the collaboration of Oriana Persico (see: Figure 1). This work is special among other implementations in the field of biometric art because—bearing in mind the nature of open-source art—it involves the real participation of internet users at various times and in different places around the world. This kind of art practice expands and redefines the usual spatial and temporal aspect of the exhibition methods of new media art. This implementation also evolved into the dimension of social campaign and performance expression, and the project also included numerous written narratives of the participants (website visitors) and speeches by the artist, as well as the initiative at the Ministry of Health in Italy which stated that the medical data of Italian citizens could be made available at their request in formats that enable free access to them all over the world (Torgovnick May 2012). According to Carsten Stage, "La Cura" project shows "( … ) how the internet and social media are technologies for both experimenting with the affective permeability and mobilizing power of body images and for (re)defining the surface of the body as the carrier of individuality during illness" (Stage 2017, p. 104). I agree that this open-source project reveals how "collaborative platforms for sharing diverse and embodied experiences of cancer" (Stage 2017, p. 104) can be created around the affective dimension of the body. For this reason, I consider the Italian artist's project to be an extremely important, subversive negotiation with the previously described mechanisms of ethopolitics. In this project, the affective

body is not only a political tool, but an object of creative and affirmative community care and attention that allows renegotiation of the status of the individual as a patient.

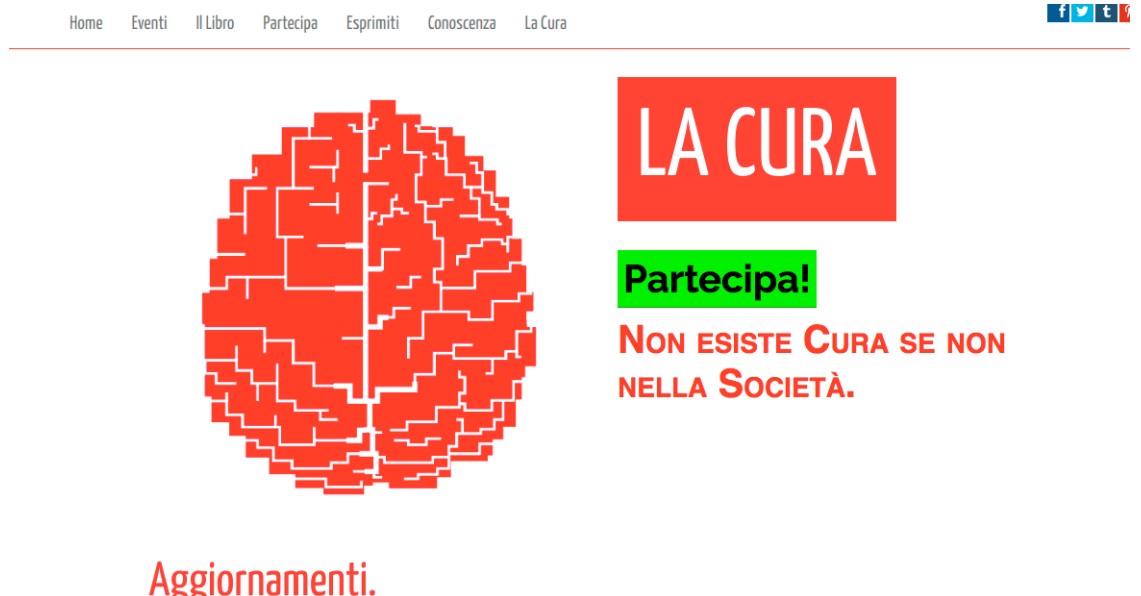

**Figure 1.** "La Cura" project screenshot from the Webpage http://la-cura.it. Courtesy of the author.

The artist's disease became part of the project material: Iaconesi was less than 40 years old when he was diagnosed with brain cancer in the form of a tumour in 2012 (See: Figure 2). The hospital in which he was diagnosed (the doctors did not offer any effective treatment for the disease) initially refused to grant him access to the documentation of his own illness. When it was finally made available to him, it was delivered in such graphic formats and described in such highly specialized language that Iaconesi could not glean anything from the data obtained. As he writes, the first stage consisted of "hacking" into the electronic files to reach the medical, verbal–visual portrait of his own illness (Torgovnick May 2013). As this Italian artist indicates in his project, the administrative management of life (or more precisely, health) which he encountered for the first time in such an intense way, regardless of whether it was carried out by the state or in the private dimension, is associated with the same degree of institutionalization: "I started to understand that this industrial process which we call medicine was not really about me. It was a reduced version of me with all the human complexity taken out of it" (Torgovnick May 2013).

Administrative and economic life management means top-down definition of certain norms and standards that are created from the perspective of private or state interests. In the case of Iaconesi's experiment, his body was disciplined by certain medical procedures, protocols and regulations. The human body was equal here to the amount of raw data obtained in the medical research procedures. As Lemke rightly pointed out, his private life was thus transformed into political and medical value related to normalized medical standards and the process of health and disease management. Therefore, for this artist this was the beginning of an extended assessment of his body's vital efficiency at administrative and institutional levels. Moreover, the imposed method of examination and treatment was to become for the artist a kind of "self-technique", a forced way of administering his own life for the coming months or years. Iaconesi was forced to adopt institutional medical guidelines as if they were his own decision on how he should optimize his health. This is how ethopolitical mechanisms work: by adapting some institutional and medical standards as a self-control strategy. As pointed out by Rose, patients who may require biological control of their own bodies also become new biopolitical entities, but this usually happens under the influence of pressure to self-optimize (Rose 2007, pp. 6–10). What is more, the artist notes that this kind of ethopolitics regime also happens at the language level,

which is why the language and narrative dimension of the project is so important: "When you have something as serious as cancer, your life disappears and you are replaced by a disease", he claims. "Doctors start speaking a language which you don't understand and which is not really meant for you to understand. Your friends and family start saying, 'What did the doctors say?' Before they even say hello. You become a disease on legs" (Torgovnick May 2013). In this way, medical language starts to create a new aspect of someone's identity—as a patient.

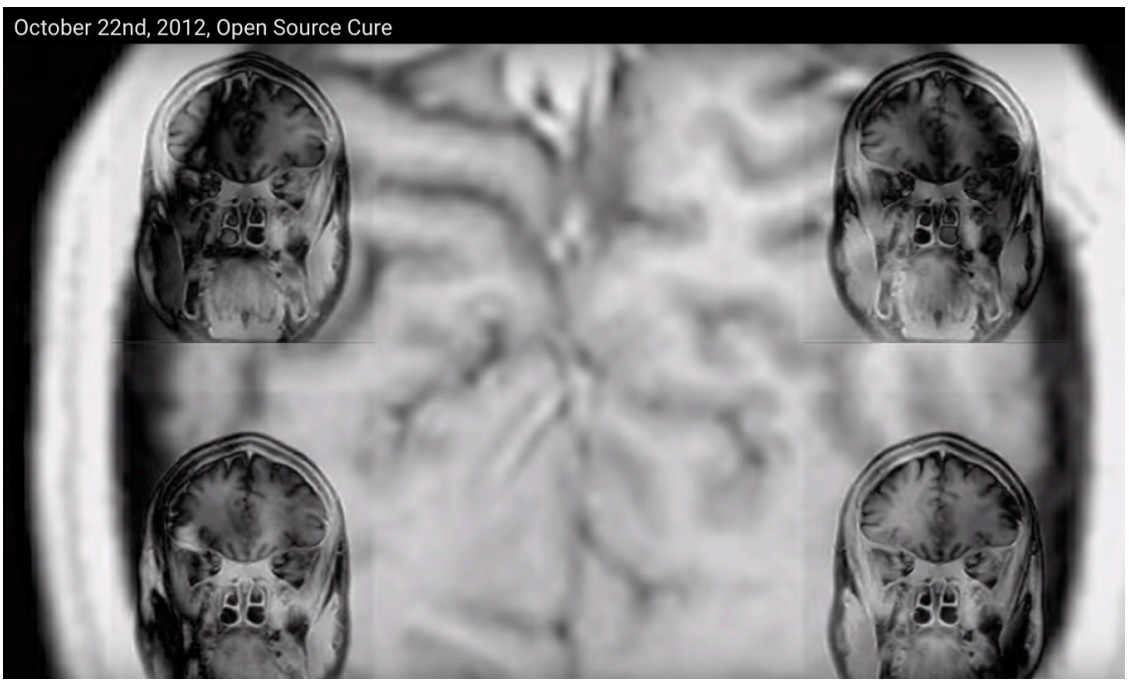

**Figure 2.** Iaconesi's brain scan screenshot from the Webpage, https://www.youtube.com/watch?v= C1C5D6pl6UM. Courtesy of the author.

For these reasons, a key element of the project was to create an internet platform that would allow Iaconesi to share all the collected documentation of his disease. In this way, "La Cura" was created as a space for the exchange of thoughts and solutions regarding the artist's illness. The idea behind the project was addressed to all internet users who would visit the site to help the artist survive the disease in any way that was accessible to them. In the first place, of course, its goal was to obtain medical expertise and to find a doctor who would perform surgery, but the implementation also had a therapeutic function in a broader scope. Iaconesi received words of consolation, advice, and also various artistic interpretations of his illness, all of which he saw as a sign of human empathy and a testimony of solidarity with his suffering (See: Figure 3). On the artistic level, Iaconesi also created a kind of transmedia forum that consisted of hybrid forms of creative expression in which verbal and non-verbal narratives met. At the narrative level, this kind of artistic activity can be seen as a special transmedia storytelling technique in relation to the definition of this phenomenon proposed by Henry Jenkins (2006). This transmedia storytelling technique took on a special form in the project because the base medium here was a scan of the artist's brain tumour, a kind of "internal photography" of illness. It became the basis for the creation of 500,000 related narratives—the responses Iaconesi received from the website (Torgovnick May 2012). Among them were poems, letters, drawings, musical works, a three-dimensional sculpture (created by Patrick Lichty), and performances which created a kind of processual staging of the disease. In this way, an unprecedented type of creative storytelling was created that was based on a collective interpretation of the artist's somatic state. The history of illness and suffering expressed initially in one photo was subsequently told by thousands of recipients, thus forming a unique transmedia platform.

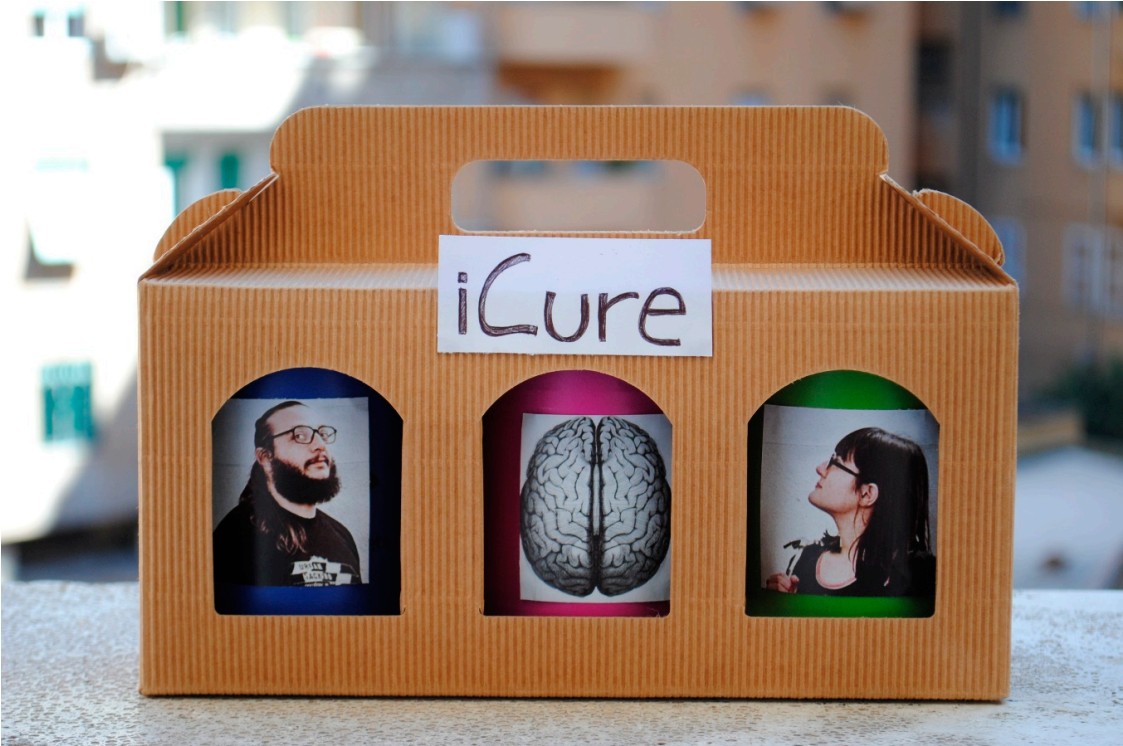

**Figure 3.** An example of community response to the project; screenshot from the Webpage
http://la-cura.it. Courtesy of the author.

This narrative experiment was also a strategy that reached beyond the administrative or economic disease management that he was offered as a part of legally sanctioned medical procedures. As a result of the involvement of specialist doctors, the most effective surgery for Iaconesi was finally developed and a successful tumour-removal procedure was performed. What is more, in 2016 a book was published in Italian in which Iaconesi and Oriana Persico described his story (Iaconesi and Persico 2016). Most of the book is written in a first-person narrative as a kind of diary of the illness and the creation of the artistic project. This diary book is a counterweight to the medical language that patients are usually surrounded by.

The artist decided to undertake this project not only in the hope of finding an effective method of combating cancer, but also as an attempt to re-empower himself as a patient. As Iaconesi emphasizes, when he learned about the disease, he suddenly ceased to feel himself and his former life was replaced by the life of the disease. His existence was instantly filtered through the prism of the tumour, which was a biological element of his body that was to some extent disconnected from the rest of his life, in relation to which further control and management of his life as an oncological patient followed (Torgovnick May 2012). Therefore, in the context of biopolitical considerations, this project can be read in two ways: as a way to cross the boarders of biopolitical institutional mechanisms, or as a kind of activity which uses the power of ethopolitics in a subversive, creative way.

Iaconesi's project primarily consisted of positioning in a different way both the disease and the treatment process, both of which were at least partially independent from administrative and economic standards and protocols. Iaconesi turned his tumour into a matter of artistic and social activity. The affective dimension of the disease became not only an object of the medical-administrative regime (Lemke 2011, p. 95), but also a non-institutional, collective platform for the exchange of thoughts and creative, affirmative potential.

From a different perspective, this Italian artist's project can be considered as ethopolitical activity (so as a part of biopolitical mechanisms), i.e., actions taken by an individual for self-optimization of health. Iaconesi took the management of his biological life into his own hands, and for this purpose he

built a procedural technical and artistic strategy of a verbal and visual nature. In this case, as Rose pointed out, the improvement of health and extension of life was not the urge to exceed limited biological abilities and to adapt to collective social rules and norms; it was an attempt to create a non-normative accompaniment in illness and recovery—a collective experience that crosses the boundary of an individual's medicalization process. As rightly noted by Stage, "Instead of telling a narrative of himself as an artist who developed cancer and thus became a patient, he—through the project—tells a different story, which is first of all able to create a strong sense of continuity between Iaconesi before and after the diagnosis" (Stage 2017, p. 107). Thus, the status of the disease and the artist as a patient changed: from the space of life and health management it was transferred to the space of collective care and social involvement in which the patient does not lose his subjective agency.

Iaconesi's project explores and diagnoses not only the biopolitical involvement of the individual in the administrative and economic mechanisms of medical procedures, but also the phenomenon of shaping the somatic identity of a person diagnosed with a deadly disease. This kind of identity, however, has no explicit pejorative connotation: it means defining oneself through various dimensions of corporeality in relation to the disease. Thanks to the strategy used by Iaconesi, every stage of shaping this identity was accompanied by artistic expression, which is a counterpoint to the naturalistic and reductionist formula to which a disease and the process of seeking a cure could generally be reduced.

## 5. Quantified-Self and "Data-Selfie": The Artistic Projects of Laurie Frick

Like transhumanism, to some extent self-tracking practices have an activist nature that is expressed most fully in the activities of Quantified-Self—a group that, due to its decentralized form, does not define itself as a "movement". This community, which is centred around the idea of the Quantified-Self, was first gathered in one place in 2008 by Gary Wolf and Kevin Kelly, editors of the magazine "Wired"; since then "Quantified-Self" meetings have been held in over 200 cities around the world, and conferences and symposia related to this new life mode are also organized (Ajana 2018, p. 2). Thus, Quantified-Self identifies communities focused on developing the idea of self-observation and improving health, as well as sharing personal experiences in this area. Wolf states that the action of Quantified-Self is based on the use of knowledge about ourselves that we can gain from the analysis of our own biological data. Therefore, in accordance with this theorist, these data are a part of our organisms and, at the same time, are objectively abstracted from the inaccessible visceral sphere so that they can constitute an important narrative cognitive source about our existence (Wolf 2016, p. 70). However, as Wolf emphasizes, these assumptions do not stem from the belief that the human being is a homogeneous, coherent "I"; on the contrary, it is a complex entity that can be analysed in various ways, including in the parameterization process, which incorporates the body into the reality of universal computation. According to Wolf, to some extent this community also anticipates the future of human beings who, with the help of appropriate devices, will soon manage their own lives, "nesting" them in reality. This will be another example of the personalization that is common in almost every area of life (Wolf 2016, pp. 68–72). Universal personalization raises various doubts which I have already pointed out, and it is similar in the case of self-parameterization techniques.

However—as is, from my perspective, most important in relation to personalization—self-tracking practices using electronic devices and software can be seen as an auto-ethnography strategy and a type of modern visual, verbal and algorithmic diary of the body. Many measurement applications provide the option of entering a comment or reflection in relation to the recorded data (for example, the You app: you-app.com), thus we can create a private verbal narrative related to health observation. It is worth emphasizing, however, that this is a specific diary that is reduced to particular words and phrases (also commands) rather than an extensive intimate story of ourselves. Regardless of the fact that these narratives are usually not as deep and complex as literary works, self-tracking diaries have another interesting aspect. Jill Walker Rettberg claims that there are generally three types of auto-presentation in the digital sphere: visual, written, and quantitative (Walker Rettberg 2014, pp. 3–4). However, she adds that users of self-tracking technologies often treat their devices and applications

not as measuring instruments but as a companion—just as people often perceived their diaries. The verbal aspect is manifested here not only in a personal narrative but also in communication with the devices (similar to "dear diary" practices). However, by subjecting the body to measurement processes or entering comments in the application, users create a kind of interactive relationship with self-trackers. Devices and applications "react" to any undertaken activity. The convention of confessing and sharing secrets gains somatic significance in self-tracking, thus becoming a tale of the internal processes occurring in the body—a kind of visceral storytelling. What is more, as Rettberg notes, some activity trackers contain a conversational agent that is based on artificial intelligence technology (such as the Lark platform) that uses broader verbal responses instead of graphical ones to comment on activity levels or tasks that should be undertaken (Walker Rettberg 2014, pp. 35–36). In this case, the personal narrative is confronted with the automatically generated narrative that is produced by the software's agency. This is especially important for chronically ill people who are looking for a confidant in their technological companion.

It seems that regardless of the ever-present biopolitical impacts of modern technologies, Braidotti's concept of "transversal alliance with technologies" that I mentioned before reflects the primary assumptions of self-tracking that are also undertaken by the activities of the Quantified-Self movement. However, the question is whether the consequences of these alliances are always positive and meaningful. Following Braidotti's observations, Neff and Nafus emphasize that self-tracking even means shifting responsibility for health and fitness to technological devices, and thus no longer transferring body management to the state or corporations but to technological non-human actors whose agility is great enough to make decisions on health issues (Nafus 2016, p. 24). This is, I think, a real threat that is arising from our growing habit of progressive automation and technicization of everyday practices. This statement confirms the reflection of self-observation made by Paolo Ruffino, who stated that the intimate engagement that is promised by self-tracker marketing campaigns never happened in his case: he also did not experience meaningful self-exploration. For him, self-tracking was only automatic data registration which did not translate into the process of constructing identity (Ruffino 2018).

Another variant of the extended hybrid narrative is shaped by applications that allow registered data to be shared with other users, as is offered by Muse (the brain sensing headband) or Endomondo. There is no room here to develop this point further, but as Rachael Kent states, "Through the sharing of self-tracking data on social media, self-monitoring of the body is extended into the communities' gaze, and the surveillance of health practices between users can be captured into digitally quantifiable as well as qualitative formats" (Kent 2018, p. 62). It is thus hard to deny that applications and the social media formula broadens the perspective of surveillance, but it is also a kind of interpersonal narrative created in relation to acquired body data. Community data observation meets the idea of Quantified-Self and creates a creative space for the exchange of thoughts on health and physical condition. Due to the mediation of specific software whose key role is a more or less friendly interface, illness or a specific ailment becomes a matter of negotiation—a process represented in the form of a visual–verbal interaction system. As a consequence, this process can generate exclusion (which is always associated with transgressing standard social norms), but in a platform of narrativization and circulation of affective knowledge it can also meaningfully re-create the sphere of alienation that is associated with disease.

Almost all aspects of self-tracking that are relevant to my topic are applied in the works of the American artist Laurie Frick. She uses various bio-tracking strategies accessible through mobile devices to create portraits—various forms of installation cycles based on data obtained from body parameterization processes. Frick, consciously using the opportunities offered by bio-tracking in the process of both self-observation and creating art, is an example of an artist and activist who is associated with the ideas of creative transhumanism. Frick's installations are not an expression of criticism of actions that strive for absolute parameterization of life; instead, they express attentive interest in transhumanist ideas of self-control, self-improvement, and self-optimization, which, according to Frick,

will next imply a change in the functioning of society. Frick says that the personal parametrizations are a picture of our hidden personality—the interior to which we do not otherwise have precise access. Moreover, although these data are most often presented in the form of repetitive diagrams and patterns, they are different for each person—they distinguish us from other people like fingerprints. From patterns and formulas that reflect certain regularities of biological functioning and the condition of the body, Frick decided to create cycles of artistic projects that take on different characters in terms of the used matter. Portraits obtained in this way are called "data selfies" by this artist (http://www.lauriefrick.com). As I mentioned earlier, they are a kind of biometric selfie. A selfie can be seen as a kind of visual self-narrative through self-creation, whereas a biometric selfie broadens this perspective by introducing generative and performative elements. Frick's auto-portraits generally take a long time to create and they point to the observed (ir)regularities. It is equally significant that the authentication mechanism is concerned here with confirmation of the factualness of some bodily experience.

Laurie Frick's works (See: Figure 4) are based on the monitoring of numerous daily activities that determine good physical condition and that can improve health: from ordinary walking activities to sleep quality and stress levels. Her "Walking", "Sleep patterns", "7 Days" or "Stress Inventory" projects should be mentioned as in these works the artist regularly monitors her everyday life activities in order to self-diagnose and improve her body condition in accordance with the idea of the "quantified-self" (http://www.lauriefrick.com). Her "Walking" work was created in 2012–2015 and was based on the FRICKbits iPhone application she had developed. Observation was made possible by motion sensors and GPS technology. Using the patterns obtained thanks to algorithms which determine the frequency and type of walking in relation to the space covered, the artist put together on a common board in the form of rectangles printed using 3D modelling. Some of them contain street and district names, which are the navigation points recorded in the process of monitoring physical activity. The number of steps performed per day affects the metabolism and keeps the body in optimal motor condition. This project could also be defined as a kind of personal cartography because it is an analysis of the existence of a bodily interior in an outer space-time—the movement of an embodied organism in the surrounding environment. When creating the project, the artist used a mobile GPS to track her daily wanderings in private and public space, thereby treating tracking of movement in space as a special diary of physical presence in the world. This is a form of a story that includes various branches in the form of personal associations. The self-tracking procedure gives biological activities a narrative character in that it makes a narrative out of everyday activities and accurately tracks their course. In the works "7 Days of a Woman 32", "7 Days of a Man 37" and "14 days", all of which were part of the installation presented in 2015 in the Pavel Zoubok Gallery, the artist compares the activities that women and men perform and measure every day: deep and REM sleep phases, the duration of walks, eating habits, heart-rate, upset stomach, etc., but also, for example, the process of waiting somewhere. All data is translated into multi-coloured squares, laser-cut charts, and graphs made of paper, and activities are coded with the colour and shape of objects (https://www.lauriefrick.com/7-days). Quite often, the elements of the installation are signed with the name of the action performed, such as "Sick", "Walk", or "Wait". The verbal elements in her works are not a story made solely for the needs of this kind of art; this is not a poetic record or even a unique form of typography, but together with visual and tactile elements it creates the quite original semiotic configurations that create the narrative structure of the project.

These data-based installations tell a new kind of story about human beings: the story of our health in the form of affective reactions that we can investigate by creating a somatic personal journal, which is also based on objective patterns. Similar to self-tracking practices in general, her art could be called a new form of auto-ethnography that is based on regular measurement of the vital functions of one's body that allows better understanding, control, and optimization of the body's condition and efficiency. Artistic creations based on abstract statistical data are extremely important here. On the one hand, self-tracking practices propose undoubtedly an implementation of the neoliberal definition of

an individual's health. As Lemke claims, determining the value (performance, productivity) of your own body by comparing the obtained data to the averaged parameters of the population increasingly means a neoliberal transfer of responsibility for life from the level of sovereign power to private entities (Lemke 2011, pp. 98–99). Interpretations of biometric data are always based on this kind of unification in reference to the statistical data of the general population. However, thanks to the fact that these data are often represented by materials such as wood, leather or aluminium, the human individual is presented as part of the physical world she/he co-creates. Therefore, data interpreted in this way cannot be treated as abstract information that is manipulated and circulated in multimodal digital systems. These are types of sculptures or special portraits, thanks to which we can confront knowledge of ourselves in a tangible way and observe the health problems of other people if they let us. In my opinion, such an artistic strategy means that affective body reactions go beyond the dataveillance space and become a real, material object of observation and experience. In an affirmative manner, Frick thus responds to the aforementioned concept of Braidotti, which shows that self-trackers can form a kind of alliance with a human being and reveals how we can and should use supportive technologies in a creative and informed way. Otherwise, bio-trackers will primarily strengthen the biopolitical idea of the "ethos of health" and cause further exclusion of sick and disabled individuals.

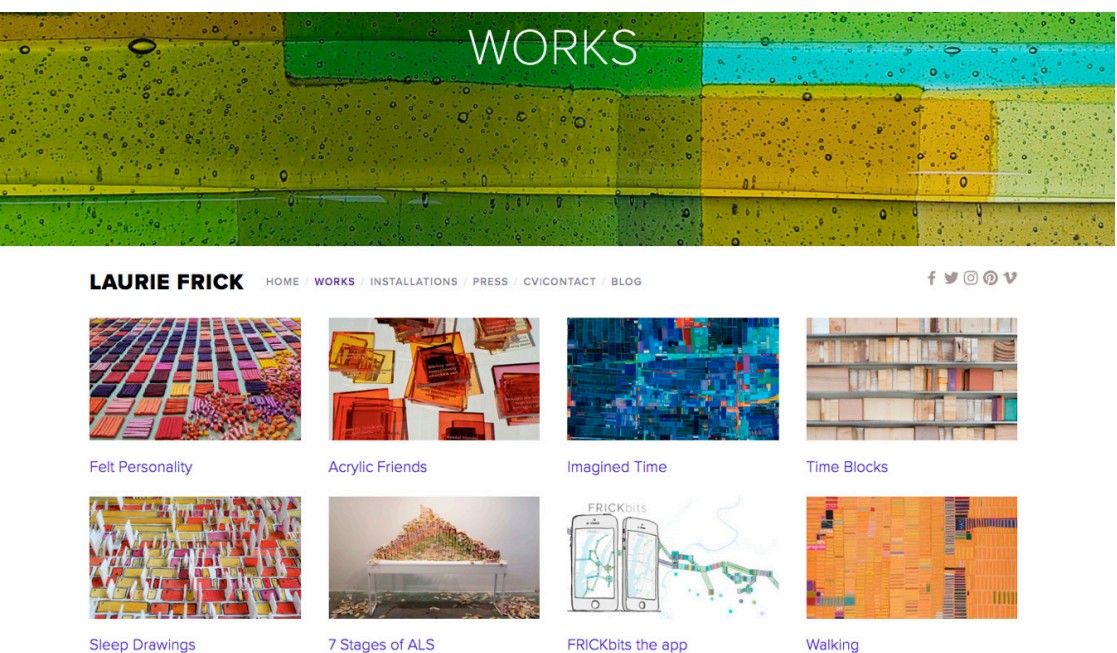

**Figure 4.** Works of Laurie Frick screenshot of the Webpage https://www.lauriefrick.com/works. Courtesy of the author.

Following the context of the assumptions of the Quantified-Self group, Frick's actions can also be determined by the processes of "re-inventing ourselves", which refer to the theory of embodied cognition that was formulated by Andy Clark in reference to the philosophy of transhumanism. Clark analyses the connections (interfaces) between people and machines that within the embodied human subject introduce self-control and reconfiguration of bodily limitations. While determining the significance of transhumanist subjectivity, he states that "( . . . ) human minds and bodies are essentially open to episodes of deep and transformative restructuring, in which new equipment (both physical and 'mental') can become quite literally incorporated into the thinking and acting systems that we identify as minds and persons" (Clark 2013, p. 113). Therefore, it is not only about exceeding the limits of the body or gaining superhuman possibilities: it is about improving and reconfiguring them, which leads to, among others, self-control and improvement. In Frick's projects, self-rediscovery is achieved by parameterizing processes of the body and monitoring its functions through available data. On the

one hand, this involves the issue of algorithmic reductionism and statistical normativization, which is conditioned by biometric procedures; on the other hand, this involves rediscovering and narrativizing corporality in relation to artistic vision.

This context is particularly important to the artist when she analyses the issue of lateral sclerosis, which is a chronic and incurable disease, in her extremely interesting project "Seven stages of ALS". ALS (amyotrophic lateral sclerosis) is a disease that causes the death of the neurons that control voluntary muscles; it is characterized by stiff muscles, muscle twitching, and gradually worsening weakness due to atrophy of the muscles. Most people eventually lose the ability to walk, use their hands, speak, swallow, and breathe (Kasugai and Machtoub 2016, pp. 1–2). Therefore, the project's goal was to symbolically attempt to regain control of this unpredictable illness by creating visible, tangible patterns (See: Figure 5)

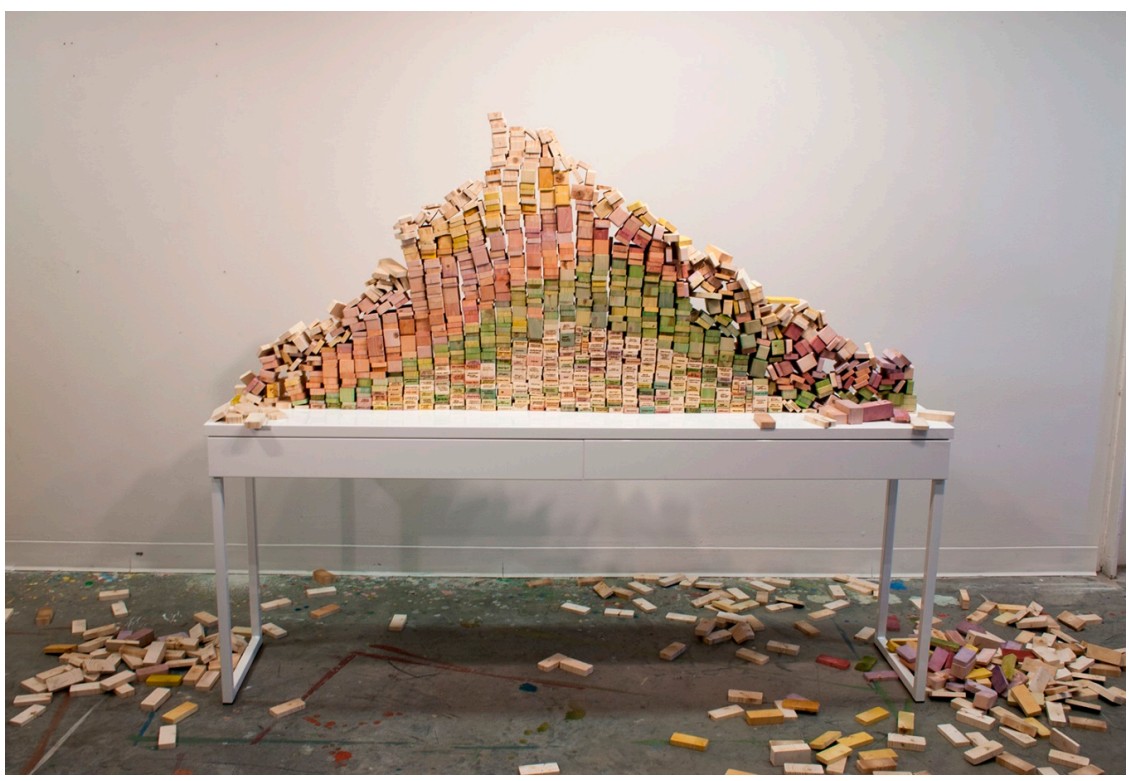

**Figure 5.** Seven Stages of ALS. Courtesy of the author.

In her installation, Frick used data (observations of walking, swallowing, etc.) from 264 patients to map out the stages of ALS. Using a series of colour-coded wooden blocks and handwritten citations, the artist was able to find patterns in the seven levels of severity seen across the six categories of ALS symptoms: (1) difficulty while walking or doing your normal daily activities; (2) tripping and falling; (3) weakness in your legs, feet or ankles; (4) hand weakness or clumsiness; (5) slurred speech or trouble swallowing; (6) muscle cramps and twitching in the arms, shoulders and tongue (Kasugai and Machtoub 2016, pp. 2–6). The installation, which was displayed at the Medical Center in Charlotte, is used not only as an emotion vehicle to bring awareness to the general public of the tragic realities of ALS, but also as a creation of awareness among patients themselves that technology can offer something other than a total loss of control in the case of illness and partial disability. A special element of the installation is hand-made inscriptions on three-dimensional constructions that are arranged in specific patterns which indicate the symptoms of the disease. These inscriptions are the occupations performed by the diseased people (architect, driver, manager, etc.) and their age in the form of a number (See: Figure 6). This is similar to a kind of database made up of language and visual data that creates a narrative specific to this type of archive: dispersed, decentralized, but related to a

specific issue, with possibilities to combine its elements in many variants (Manovich 2002, pp. 199–200). Similar to Frick's other projects, handwritten terms gain personal character and create specific kinds of verbal–visual representations whose goal is to show that serious illness is not synonymous with losing other parts of one's identity. Illness is here rather integrated with self, occupation, age and gender, and it is not the only factor that defines the identity of even a suffering individual. What is more, these verbal incrustations also clearly show that ALS can affect any person, regardless of their age and occupation, which is why the installation becomes a kind of core condensation of disease in a form of a hybrid sculpture-collage which is told and shown from the perspective of the patients. (https://www.lauriefrick.com/7-stages).

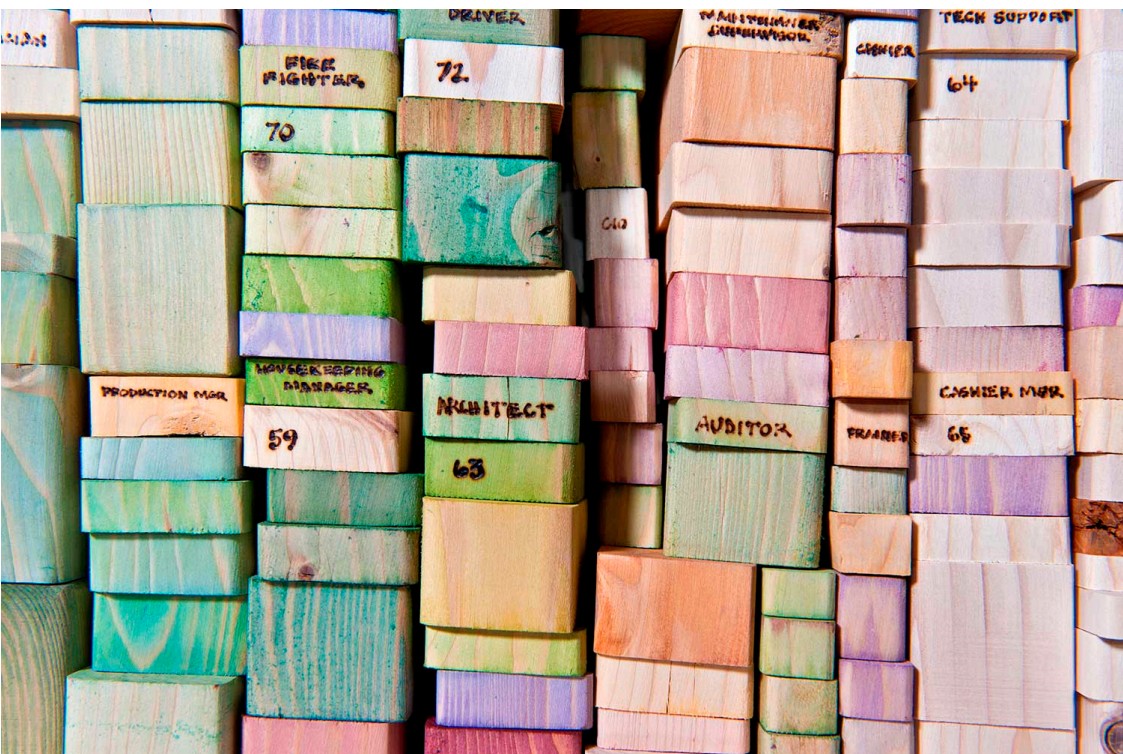

**Figure 6.** Seven Stages of ALS. Courtesy of the author.

The Frick installation basically does not consist of photographs, but it is presented to a wider audience as photographic documentation of this artistic activity, what is very often crucial element of many kinds of conceptual art. It is not the primary form of exposition, but certainly the main one in digital space. The data in the project are therefore double visually mediated: first by the wooden elements, secondly by the medium of photography. Photography is a very important element of the whole here, because as I wrote earlier, Frick creates her works as types of "data-selfie", pictures that represent biological data of the measured body.

I think that this kind of materialisation of ALS's quite ambiguous symptoms seems closer to the perception and feeling of health and illness in general. The cultural representation of this illness acquires here an interesting medical-related dimension in which medical technologies are combined with the visual and verbal design and lifestyle patterns. In this aspect of the project, the ethical power of its influence is also revealed. However, this is not about simply creating compassion but about making ALS patients visible in society as professionally active people whose disease can to some extent be managed by technology. The artist's project illustrates the agential possibilities of diseased and disabled people, who should not be perceived as passive objects of our simple empathy but as active subjects with whom we create various relationships and in whose lives we can get involved. As some research shows (Hofmann and Fredrik 2018), self-tracking practices can change the perception

of disease and sick people, who gain simple quasi-multimedia tools with which they can control their condition. Despite the problem of biological control and observation due to the micropolitics of large concerns, it seems that diseased people gain some autonomy thanks to self-tracking.

Frick's project, like the others I have chosen, not only illustrates technological procedures in the context of reflection on health and disease, but also nuances this field of reflection significantly, pointing to the changes that are made through such kind of nonlinear narratives based on technological solutions in reference to the situation of diseased or disabled people. Disease not only creates a space of Otherness (as in neoliberal ethics), but (just like in Iaconesi's project) it can be a forum for the exchange of thoughts as well as an incentive to carry out real social change. In Frick's projects, people who are diseased or who just monitor their health are not shown as victims or excluded others; within a specific community they can become agential subjects that have a real impact on their own health and functioning in the world. Photography and other forms of visual data representations are for them additional tools to regain health control. We, as recipients of the project, should not only sympathize with these people but see their possibilities and limitations just as we observe our own—in reference to obtained data. Technology which leads to standardization and statistical unification thus in this case has a clear subversive meaning.

## 6. Conclusions

Knowledge systems related to the biological dimension of the population produce various normative mechanisms to which the authorities grant themselves the right to determine the value or worthlessness of life (in a social and research context). These norms and regulations indicate the elements that threaten or support health and life and define forms of social and cultural exclusion, not only in relation to the problem of race or ethnicity, but also to the physical and mental condition of individuals. Subjectivity perceived in this dimension or in terms of an information base is becoming the subject of numerous political operations performed in the medical, religious, moral, and scientific fields. As Lemke notes in his conceptualization of contemporary biopolitical activities, the following questions are now becoming some of the most important:

> "What existential hardships, what physical and psychic suffering, attract political, medical, scientific, and social attention and are regarded as intolerable and as a priority for research and in need of therapy, and which are neglected or ignored? How are forms of domination, mechanisms of exclusion, and the experience of racism and sexism inscribed into the body, and how do they alter it in terms of its physical appearance, state of health, and life expectancy? ( . . . ) Who bears the costs and suffers such burdens as poverty, illness, and premature death because of these processes?". (Lemke 2011, pp. 119–20)

The discourse investigated by this philosopher covers one of the most important issues of biopolitics, which can be generally described as the "ethos of health" in relation to artistic practices based on biometrics. These regularities are particularly evident in phenomena associated with biological parametrization processes, i.e., quantification of the body's functions that illustrates their vitality and identifies health threats. In addition to parametric identification as a method of ensuring population safety, this ethos is part of the "care" paradigm that was described by Foucault and which within his assumptions has a negotiating, ambiguous status related to biopolitical procedures (Foucault 1986). Concern for broadly understood security (territorial, health, economic, etc.) in this era of increasingly complex globalization processes is on the one hand authorities' most important obligation to society. On the other hand, it inevitably involves a sphere of influence in which politics is identical to economics.

In relation to biopolitical processes, the projects in the area of art and science discussed in this article in various ways examine the issues of health, illness, and the vitality of the body. Developing individual narratives in a transmedia manner, they use verbal and visual codes to diagnose the contemporary subjectivity of the diseased person and to point to various aspects of the biomedicalization process that patients deal with. The use of medical imaging and various forms of biometric visualizations in artistic

projects makes it possible to re-contextualize medical procedures related to the diagnosis and treatment of disease; biometric data are incorporated in individual narratives that establish a more universal and subversive representation of what we can define today as a disease or, more precisely, a state of being unhealthy. Medical imaging and biometric visualisation changes the traditional meaning of artistic and cultural reflection on health; by using photography, forms of diaries, collages, or forums for the exchange of thoughts, the artist presented in this article introduced an aspect that is often overlooked in visual narratives: data from inside the body.

This article has presented three different variants of the creation of biomedical stories about disease from the perspective of data obtained from inside the body thanks to the possibilities provided by modern medical devices. In projects based on biometric data, the indicated imaging methods are usually one of the elements of the entire artistic work, so creative concepts are not limited to the use of biomedical procedures alone. In all projects, the most important issue is the context in which the projects were created and the problems they illustrate. Verbal–visual collages depicting damaged, suffering bodies as a kind of affective weapon in the processes of media globalization; the implementation of open-source art as a platform for the exchange of thoughts and social debate on cancer; self-tracking as a form of chronic disease control and a subversive biopolitical tool: these are just some of the new situations related to disease that contemporary artistic reflection examines. They are also excellent examples of work in which modern technological solutions—and forms of photography—are often associated with the most intimate reflection on disease; they show that these areas are not as distant as we used to think; they have positive and negative consequences.

As the artists I refer to show, "transversal alliances with technologies" also relate to spaces that are most taboo in reflection on contemporary identities. These kind of untold stories gain new forms of expression through the links between art, science and technology. As a sphere of affective reactions entangled in a variety of biopolitical mechanisms, (vulnerable) body becomes a political tool in two ways: as an element of ideological discourses, but also as a set of creative strategies that give bodily policy a subversive meaning. Health and disease can be defined in this context not as a certain established state of the body, but as a dynamic construction of transforming body, which is subjected to both discursive and administrative normativization processes, as well as creative practices in which immanent body agency is explored. Modern biometric techniques enable tracking of these bodily changes while constructing a strong mechanism for the somatic control of individuals.

**Funding:** This article is the result of the research project No. 2014/15/N/HS2/03926 which has been funded by the National Science Centre (Preludium 8) for the years 2015–2019.

**Conflicts of Interest:** The authors declare no conflict of interest.

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
