# Peer review of "Somatic Narratives about Illness. Biometric Visualization of Diseased and Disabled Bodies in Art and Science Projects"

_humanities, doi:10.3390/h9010019_

Round 1
Reviewer 1 Report
The exploration of artistic practices around the datification of health is interesting. However, the article lacks a consistent critical critical engagement with the production of knowledge around the body, health and care. Foucault's biopolitics and Lemke's biopolitical society are mentioned in the abstract, however, the text is not articulated through this approach, not it is well explained why it is relevant for the analysis of the artworks/artistic practices. In summary, the article needs a much more critical engagement with the subject matter. From Wilson's informational arts to physiognomy. What ideological discourses were sustained and reaffirmed through physiognomy? Why is physiognomy relevant today? What discursive formations link physiognomy in the past with physiognomy in the present? The articulation of the body as a political tool needs to be clearer, more substantial and be placed at the start of the text. There is much description and little analysis and criticism of ideology and discourse.
The discursive categories discussed through the exploration of concrete artworks need to be presented in a systematic, organised and clear manner at the start of the text. The reader would then understand the body of literature, and thus the theoretical framework employed for the analysis. These categories should then be reviewed at the end in a conclusion that goes beyond the analysed artworks and perhaps contextualises artistic practice through/with body visualisation within the discourse of austerity, class, health and the dismantlement of the welfare state.
The rationale for employing theory of affect for the exploration of the selected artworks/practitioners needs to be much more succinct. There is some repetition and a lot of unnecessary description. The work of Clough and Brennan is reviewed with certain level of detail but it appears disconnected from the aims and objectives of the article. Key terms need to be briefly presented and then contextualised within the case study, namely the selected art practice.
The article would gain clarity with the inclusion of some images of the artworks under analysis.
In lines 62 to 64 the author explains that body imagining such as xrays are combined with other semiotic representations in the context of art, however, this is not specific of art. These visualisations often contain some linguistic material, and are shared among practitioners of medicine and patients/the subjects whose bodies are being render visible, alongside verbal language.
Often, key terms are not well situated within literature. This is the case for "quantified self" in line 114, and corporeality in line 122.
The link between Hans Belting's medium-image-body and Diane Covert's project needs to be unpacked. For someone who is not familiar with Belting's work, it won't make sense. However, a literature review that weaves Foucault's biopolitics with Belting's triad and affect theory as a framework of analysis for the type of art the author explores, would be of great interest to media researchers, human geographers, art historians, and anthropologists alike.
Minor spelling, grammar and style mistakes throughout the article. For example in abstract lines 7 and 8, is it "I will analyze, and use specific medical tools", or "I will analyze the use(s) of".
Author Response
Answer to the review No. 1
First of all, I would like to thank the Reviewer for an comprehensive and precise review, thanks to which I could correct and improve my article. In my answer, I will try to be equally comprehensive to each of the Reviewer’s comments.
I agree, that the article lacks a consistent critical engagement with the production of knowledge around the body, health and care. Therefore I developed a part of the text concerning the analysis of discourse and ideologies related to the ancient and modern techniques of body parameterization, and I described the consequences of these techniques. I also tried to show that the projects I analyze deal with this discourse, but artistic expression also serves to indicate subversive applications of body mapping and parameterizing technologies. I hope that thanks to this elaborations I introduced consistent critical engagement with the issue of body, disease, health and care.
In reference to Reviewer’s suggestions, I developed Thomas Lemke's theory so that it becomes a useful conceptual apparatus in the later analyzes of artistic projects. I have also refined the passage on the concept of Nikolas Rose. I have considerably shortened the fragment recapitulating the determinants of affective theory, extracting only those categories and descriptions of phenomena that I use in subsequent analyzes. I focus primarily on the body's involvement in biopolitical mechanisms and the activities of modern media technologies. This topic is a fundamental point of reference to how health and illness are conceptualized in the artistic projects I have chosen. What is more, I tried to presented discursive categories discussed through the exploration of concrete artworks in a systematic, organized and clear manner at the start of the text – in the first two subchapter of the paper.
I agree, that some semiotic elements (especially verbal ones) are not specific to this type of art, but it should be emphasized that in this form they are also not characteristic for the description of medical images. On x-ray or tomographic images, hermetic, used medical language describes anatomic pathologies, often with the help of abbreviations or established medical markings. In Covert’s, Frick’s, and even more Iaconesi's projects, the verbal layer definitely are not identical with the standard of medical description. I have clarified this context in my text.
I also agree that the article would gain clarity with the inclusion of some images of the artworks under analysis. Therefore, I contacted two of the artists. They have agreed to use the photos and are also very interested in my considerations.
Thank you for comment that “Foucault's biopolitics with Belting's triad and affect theory as a framework of analysis for the type of art the author explores, would be of great interest to media researchers, human geographers, art historians, and anthropologists alike”. I definitely agree, but there was no place for this considerations in my article. Belting’s concept is very complex, and I would like to rather develop the parts focused on artistic projects analyses. I will try to do this in the next paper, showing the connections between body, image and medium from Belting’s theory in reference to the biopolitics and affect theory. In this case I delated the part about Belting theory.
I am aware that the article should be proofread – the revised version of it was corrected by UK native speaker.
Finally, I would like to thank again for careful reading of my text.
Reviewer 2 Report
The article is interesting within a growing field of art-science projects. I have the following suggestions for you in order to clarify your endeavour:
A clearer distinction could be made between the theoretical framework and the analytical paragraphs. Sometimes the paper tends to have exclusively a theoretical focus with the art works as exemplifying what the theories put forward. I recommend you to make a clearer distinction between the theoretical framework and your readings of the art works. The referencing of others' work tend to continue in the analytical part of the article which makes it difficult to extract your analytical points from what the scholars you are citing are saying about the topic in general. Give us some methodological considerations: Why this three cases? Argue for your choice and how do you read them? (Important concepts for example) You do have some analytical points - in particular around the Quantified Self - you could make clearer your points in the two other cases. Stress those in the conclusion. I know the following is a matter of tradition, but I have huge difficulties with understanding how it is possible to work with "affective contagion", and the affective experience of audiences without ever taking them and their reactions into consideration as an empirical fact? All we can say is that the art work puts out cues that tend to attune audiences (Massumi 2008), but as researchers we have to document that this is what is going on (participant observations., interviews, creative methods, or at least auto-ethnography). This has huge impact on the reading of the work Inside terrorism because we have to do here with body-to-body-relationships stripped of racial, ethnic, political, social and subjective features and I wonder whether the affective experience would happen. at all?? It has to be documented, in my mind. I am not a native speaker but there are quite a lot of language issues in the paper.Author Response
Answer to the review No. 2
First of all, I would like to thank the Reviewer for a comprehensive and precise review, thanks to which I could correct and improve my article. In my answer, I will try to be equally comprehensive to each of the Reviewer’s comments.
I agree, that a clearer distinction making between the theoretical framework and the analytical paragraphs was necessary for my paper. I introduced a clearer division between these parts, supplementing the text with an additional subchapter in which I briefly describe my theoretical base. First of all, in this section I deal with the issue of the body as a political tool (as recommended by the Reviewer). I show this problem in relation to the theory of affect and the concept of biopolitics in the view of Thomas Lemke. I develop Lemke's theories so that it becomes a useful conceptual apparatus in the later analyzes of artistic projects. I have considerably shortened the fragment recapitulating the determinants of affective theory, extracting only those categories and descriptions of phenomena that I use in subsequent analyzes. I focus primarily on the body's involvement in biopolitical mechanisms and the activities of modern media technologies. This topic is a fundamental point of reference to how health and illness are conceptualized in the artistic projects I have chosen.
In the following subsections, I developed the methodological perspective, but I announced each of the categories used, when I presented my methodology. What's more, I made clearer points in all the cases of artistic projects I analyzed. Reviewer’s comments helped me to better organize the text and make it more accessible to the reader.
For me, the most problematic part of the review is the fragment about the use of affect theory in the description of mediated works of art. As the Reviewer (her)himself mentioned, it is a matter of a certain interpretative tradition. Therefore, it is difficult for me to convince someone for a research perspective for whom this perspective is not useful. A lot of books have been written (as Laura Podlasky’s “The Politics of Affect and Emotion in Contemporary Latin American Cinema”, Paul Gromley’s “The New-Brutality Film. Race and Affect in Contemporary Hollywood Cinema”, Carsten Stage’s, “Networked Cancer. Affect, Narrative and Measurement” to mention just some of many examples) in which art (from photographs to performances) is analyzed in the context of affect theory without including empirical research. Massumi himself definitely develops this issue in his book Semblance and Event. Activist Philosophy and the Occurrent Arts (2011). What's more, similar considerations have also for a long time been related to performativity, which, according to many researchers, can be implemented in the mediatized form (Auslander 2008). I conduct some empirical research among my students (using biometric devices, for example MUSE band and Affectiva - the invention of Rosalind W. Picard). Based on this research, I could establish that it is very difficult to translate the results into very specific conclusions about affective experiences (equipment limitations and often no direct translations to felt emotions) and often does not bring much more than the statement that intensive affective reception takes place in response for works of art that are based on representations of diseased body. I strongly respect the opinion of the Reviewer, and partially I agree, but the purpose of my text was not to present empirical research (I want to do it elsewhere in the article on another topic), but to show different bodily variants of disease and health representation in artistic interpretation.
I am aware that the text required proofreading - I even said it to the editor of this special issue. The article has already been reviewed and corrected by a native UK speaker (in UK spelling).